# Intrinsically aggregation-prone proteins form amyloid-like aggregates and contribute to tissue aging in *Caenorhabditis elegans*

Chaolie Huang[1], Sara Wagner-Valladolid[2†], Amberley D Stephens[2†], Raimund Jung[1], Chetan Poudel[2], Tessa Sinnige[3], Marie C Lechler[1,4], Nicole Schlörit[1,4], Meng Lu[2], Romain F Laine[2‡], Claire H Michel[2], Michele Vendruscolo[3], Clemens F Kaminski[2], Gabriele S Kaminski Schierle[2*], Della C David[1*]

[1]German Center for Neurodegenerative Diseases (DZNE), Tübingen, Germany; [2]Department of Chemical Engineering and Biotechnology, University of Cambridge, Cambridge, United Kingdom; [3]Department of Chemistry, University of Cambridge, Cambridge, United Kingdom; [4]Graduate Training Centre of Neuroscience, University of Tübingen, Tübingen, Germany

*For correspondence:
gsk20@cam.ac.uk (GSKS);
della.david@dzne.de (DCD)

†These authors contributed equally to this work

Present address: ‡Medical Research Council Laboratory for Molecular Cell Biology, University College London, London, United Kingdom

Competing interests: The authors declare that no competing interests exist.

**Abstract** Reduced protein homeostasis leading to increased protein instability is a common molecular feature of aging, but it remains unclear whether this is a cause or consequence of the aging process. In neurodegenerative diseases and other amyloidoses, specific proteins self-assemble into amyloid fibrils and accumulate as pathological aggregates in different tissues. More recently, widespread protein aggregation has been described during normal aging. Until now, an extensive characterization of the nature of age-dependent protein aggregation has been lacking. Here, we show that age-dependent aggregates are rapidly formed by newly synthesized proteins and have an amyloid-like structure resembling that of protein aggregates observed in disease. We then demonstrate that age-dependent protein aggregation accelerates the functional decline of different tissues in *C. elegans*. Together, these findings imply that amyloid-like aggregates contribute to the aging process and therefore could be important targets for strategies designed to maintain physiological functions in the late stages of life.
DOI: https://doi.org/10.7554/eLife.43059.001

## Introduction

Aging is a gradual decline in physiological functions and organ integrity and indeed diminished physical capacity and cognitive functions are already apparent before midlife (*Belsky et al., 2015*). The aging process greatly enhances the risk for chronic diseases leading to long-term disabilities in the elderly population and often premature death (*Kaeberlein, 2013*). A better understanding of what drives aging and in particular functional decline holds the promise of identifying targets to maintain quality of life with old age. It is generally agreed that the ultimate root causes of aging occur at the molecular level (*Kaeberlein, 2013*). One of the most universal hallmarks of aging is increased protein instability (*Golubev et al., 2017*; *Gorisse et al., 2016*). In young individuals, an efficient protein homeostasis (proteostasis) network prevents the accumulation of damaged proteins. However, overwhelming evidence points to a collapse in the proteostasis network with age and consequently a decline in the ability to cope with protein physical and chemical instability (*Taylor and Dillin, 2011*).

Still the role of protein instability in aging is unclear (*Finch and Crimmins, 2016*; *Hekimi et al., 2011*).

During aging, an important cause of protein instability is cumulative damage through non-enzymatic posttranslational modifications occurring through reactions with metabolites and reactive oxygen species (*Golubev et al., 2017*). Destabilizing mutations, transcriptional and translational inefficacy are also a significant source of protein instability (*Nedialkova and Leidel, 2015*; *Vermulst et al., 2015*). In a disease context, a specific form of protein instability, namely protein aggregation, is a common feature of amyloidoses and many neurodegenerative diseases. In these diseases, a specific protein converts from its native soluble conformation into an insoluble state by self-assembly into a cross-β sheet filamentous structure termed amyloid fibril. Amyloid fibrils accumulate as pathological deposits in a variety of different tissues. Although most proteins contain amyloid-promoting sequences and form amyloid fibrils in appropriate conditions in vitro (*Chiti et al., 1999*; *Goldschmidt et al., 2010*), there is little evidence for this disease-related conformational state in metazoa in the absence of disease. Global proteomic characterizations of age-dependent protein insolubility, a hallmark of disease-associated protein aggregation, have been carried out in wild-type *C. elegans* (*David et al., 2010*; *Reis-Rodrigues et al., 2012*; *Walther et al., 2015*). Several hundred proteins enriched in distinct structural and functional characteristics were identified that lose their native structure in aged animals and become highly insoluble in strong detergents (*David et al., 2010*; *Lechler et al., 2017*; *Reis-Rodrigues et al., 2012*). In vivo data demonstrate that these proteins accumulate in solid/immobile structures in aged animals. However, until now it is not known whether these age-dependent protein aggregates contain amyloid fibrils. Elucidating the structural nature of this novel type of protein instability would help to understand the causes and consequences of protein aggregation during aging and to explain potential interactions with disease-aggregating proteins.

Since the initial discovery in *C. elegans*, age-dependent protein aggregation has been demonstrated in different organs in mammals (*Ayyadevara et al., 2016a*; *Ayyadevara et al., 2016b*; *Groh et al., 2017a*; *Leeman et al., 2018*; *Tanase et al., 2016*). Importantly, age-dependent protein aggregation is likely to be relevant in neurodegenerative processes. Indeed, the over-representation of proteins prone to aggregate with age among proteins sequestered in Alzheimer's disease inclusions of amyloid plaques and tau neurofibrillary tangles highlights a possible contribution to pathological processes (*Ciryam et al., 2013*; *David et al., 2010*). Moreover, recent evidence shows that aggregates from wild-type aged mouse brains are a potent heterologous seed for amyloid-β (Aβ) aggregation (*Groh et al., 2017a*). However, because of the difficulties to distinguish the effects of protein aggregation from other consequences of aging, it is still not clear whether age-dependent protein aggregation plays a role in accelerating the aging process (*David, 2012*). A previous study has shown that *C. elegans* treated with RNAi targeting genes encoding aggregation-prone proteins tend to live longer than randomly chosen targets (*Reis-Rodrigues et al., 2012*). However, the interpretation of these results is complicated by the loss-of-function of these proteins. Recently, we found that aged animals with high aggregation levels of an RNA-binding protein (RBP) with a low-complexity prion-like domain were shorter lived, significantly smaller and less motile than animals with low RBP aggregation levels (*Lechler et al., 2017*). Still this evidence does not provide a definite answer to whether age-dependent protein aggregation plays a causal role in aging rather than being a simple byproduct. To understand whether protein aggregation is protective or harmful, it is informative to look at how long-lived animals modulate protein solubility. Longevity mechanisms and enhanced proteostasis are tightly coupled (*Taylor and Dillin, 2011*). However, whereas several studies show reduced age-dependent protein aggregation in long-lived animals (*David et al., 2010*; *Demontis and Perrimon, 2010*; *Lechler et al., 2017*), a recent study suggests that enhancing protein aggregation could be a strategy to promote longevity (*Walther et al., 2015*).

In the current study, we use a combination of transmission electron microscopy (TEM), fluorescence lifetime imaging (FLIM) (*Kaminski Schierle et al., 2011*), Thioflavin T (ThT) staining and structured illumination microscopy (SIM) (*Young et al., 2016*) to reveal amyloid-like structures in age-dependent protein aggregates in vivo. Unlike protein chemical instability caused by cumulative damage, age-dependent protein aggregates are formed by intrinsically aggregation-prone proteins, even shortly after their synthesis. We demonstrate that age-dependent protein aggregation is toxic and contributes to functional decline in *C. elegans*.

## Results

### Age-dependent aggregating proteins are intrinsically prone to aggregate in certain tissues

Previously, we performed an extensive characterization of the aggregation of two proteins, casein kinase I isoform alpha (KIN-19) and Ras-like GTP-binding protein rhoA (RHO-1). Both KIN-19 and RHO-1 were identified among the proteins with the highest propensity to become insoluble with age in wild-type *C. elegans* somatic tissues (*David et al., 2010*). In vivo analysis of animals expressing these proteins fused to fluorescent tags showed the appearance of immobile deposits with age (*David et al., 2010*). Among the insoluble proteome, the enrichment of certain physico-chemical features such as high aliphatic amino acid content or propensity to form β-sheet-rich structures shows that age-dependent protein aggregation is not random (*David et al., 2010*; *Lechler et al., 2017*; *Walther et al., 2015*). To understand whether KIN-19 and RHO-1 have an intrinsic capacity to aggregate similar to disease-associated proteins or whether a progressive accumulation of protein damage caused by non-enzymatic posttranslational modifications is required to induce their aggregation, we evaluated the dynamics of protein aggregation in vivo. Protein labeling with mEOS2, a green-to-red photoconvertible fluorescent protein, has been successfully used to track protein dynamics (*McKinney et al., 2009*). In the present case, we used the mEOS2 tag to investigate how fast newly synthesized KIN-19 and RHO-1 aggregate. For this purpose, we generated transgenic animals expressing KIN-19::mEOS2 in either the pharynx or in the body-wall muscles and transgenic animals expressing RHO-1::mEOS2 in the pharynx. The mEOS2 tag did not disrupt the aggregation potential of KIN-19, as the absence of fluorescence recovery after photobleaching confirms that both KIN-19::mEOS2 puncta in the pharynx and body-wall muscle are highly immobile structures (*Figure 1—figure supplement 1A–D*).

To follow newly synthesized proteins, we set-up a system to perform irreversible photoconversion of the mEOS2 tag present in live animals from green to red by exposing them to intense blue light. At a defined time-point, we photoconverted the mEOS present in aggregates to red. After the photoconversion, newly synthesized proteins emitted green fluorescence and could thus easily be distinguished from old (photoconverted/red) aggregates. This method allowed us to follow the rate of new aggregate formation and the rate of old aggregate removal in a population of transgenic animals over time. We observed photoconversion of all aggregates, but we also noted that the core region of some larger aggregates continued to emit green fluorescence (*Figure 1A,B*). Interestingly, we observed a doubling in the number of animals with newly-formed green aggregates, 24 hr after the conversion at day 5 of adulthood in both the pharyngeal and body-wall muscles. Conversely, analysis of the red aggregates suggested a slow removal of old aggregates (*Figure 1A,B*, *Source data 1*). Blocking translation shortly before conversion reduced the formation of new aggregates (*Figure 1B*). Conversion in young animals and blocking translation produced similar results (*Figure 1—figure supplement 1E*). Confocal imaging shows that newly synthesized KIN-19 associate with pre-existing aggregates (*Figure 1C*). However, we also observed large aggregates emitting only green fluorescence suggesting that seeding is not a prerequisite for KIN-19 aggregate formation (*Figure 1C*). We observed similar aggregation kinetics in animals with RHO-1::mEOS2 expressed in the pharynx (*Figure 1—figure supplement 1F*). Together, these results reveal that aggregate formation can proceed rapidly after protein synthesis.

### Age-dependent protein aggregates contain amyloid-like structures in vivo

Both KIN-19 and RHO-1 are nearly identical to their human orthologues (sequence identity over 87%). X-ray diffraction shows that the human orthologues are globular proteins formed by a series of α-helices and β-strands. The ability of KIN-19 and RHO-1 to aggregate shortly after synthesis suggests that their aggregation may be associated with the presence of folding intermediates. Notably, both proteins have several segments with a high amyloid propensity (*Figure 2—figure supplement 1A,B*) (*Goldschmidt et al., 2010*). These features raise the possibility that KIN-19 and RHO-1 gain an amyloid-like conformation during their aggregation with age. We have previously shown that FLIM can be used to determine whether certain proteins are likely to form amyloid-like aggregates as the formation of amyloid fibrils leads to a significant drop in the fluorescence lifetime of the

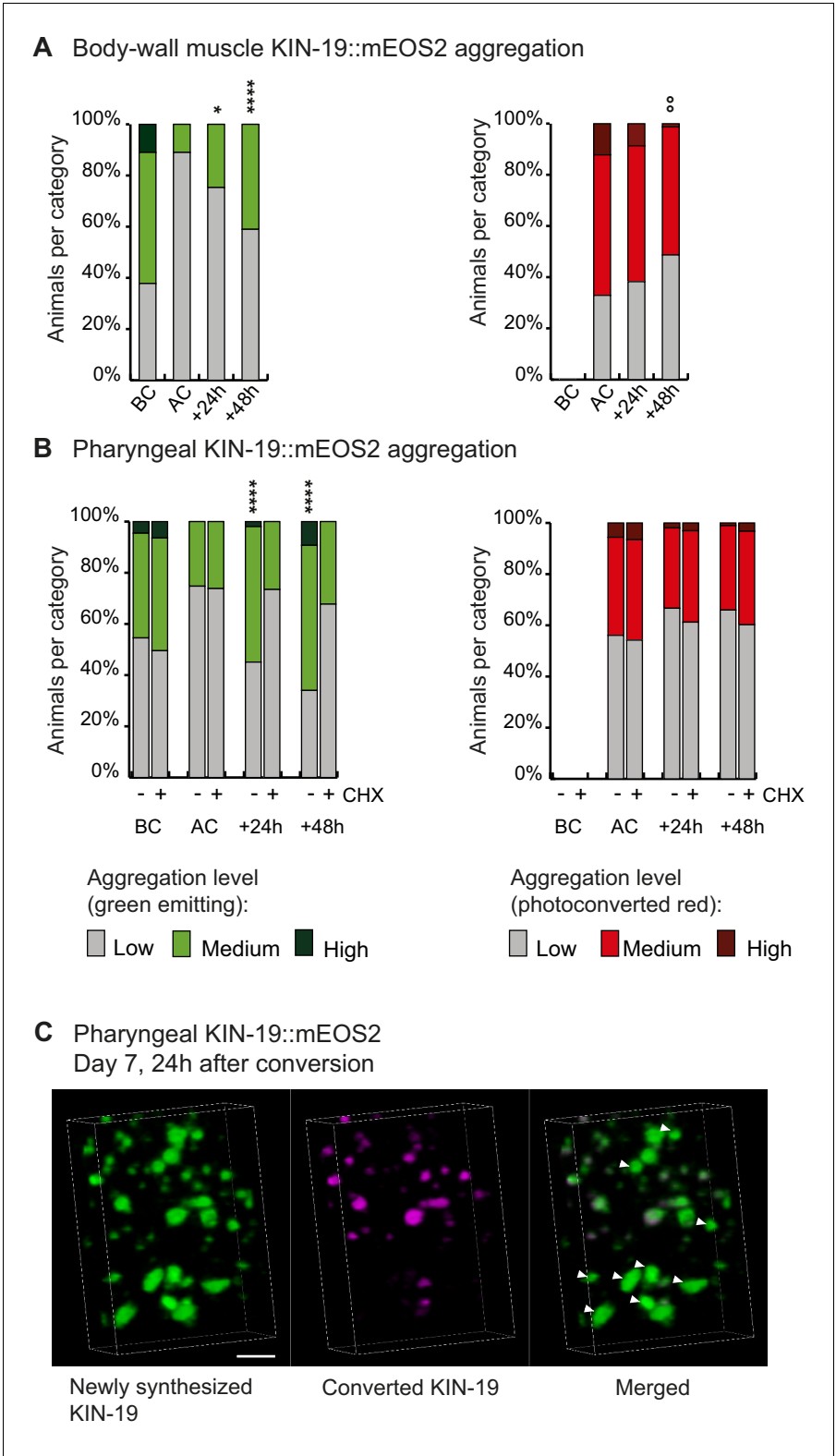

**Figure 1.** Newly synthesized KIN-19 rapidly transitions into aggregates in aged animals. (**A**, **B**) Following photoconversion at day 5 in the pharynx or in the body-wall muscle, the number of animals with newly synthesized green-emitting non-converted KIN-19::mEOS2 aggregates doubles over 24 hr. Conversely levels of red-emitting converted aggregates slowly declines. Blocking translation with cycloheximide (CHX) prevents new aggregate formation (**B**). Aggregation is evaluated in *Pkin-19::KIN-19::mEOS2* and *Pmyo-3::KIN-19::mEOS2* transgenic animals

*Figure 1 continued on next page*

*Figure 1 continued*

on the basis of the fluorescence intensity in puncta, representing aggregation (see Materials and methods). Quantification was done in the same population of *C. elegans* before conversion (BC), after conversion (AC), 24 hr after conversion and 48 hr after conversion. Fisher's exact test comparing low versus medium +high aggregation levels to after conversion: *p<0.05, **p<0.01, ****p<0.0001. Fisher's exact test comparing low +medium versus high aggregation levels to after conversion: °p<0.01. See source data including independent biological repeat in *Source data 1*. (C) 24 hr after photoconversion at day 7, newly synthesized KIN-19::mEOS2 (green emitting) forms new aggregates and associates around older aggregates (red emitting). 3D reconstruction in the pharyngeal anterior bulb region. Arrow heads highlight large new aggregates formed independently of previous aggregates. Scale bar 2 μm.

DOI: https://doi.org/10.7554/eLife.43059.002

The following figure supplement is available for figure 1:

**Figure supplement 1.** Newly synthesized KIN-19 and RHO-1 rapidly transitions into aggregates in young animals.
DOI: https://doi.org/10.7554/eLife.43059.003

fluorescent-tagged amyloid proteins due to quenching (*Chen et al., 2017*; *Kaminski Schierle et al., 2011*; *Murakami et al., 2015*). In the current study, we applied FLIM to determine whether KIN-19 and RHO-1 form amyloid-like aggregates in live *C. elegans*. For this, we generated transgenic animals expressing translational fusions with the yellow fluorescent protein Venus in the pharynx. We first confirmed that there is an age-dependent increase in aggregate formation by KIN-19::Venus and RHO-1::Venus in the pharynx (*Figure 2—figure supplement 2A,B*) (*Lechler et al., 2017*). Using FLIM, we found that pharyngeal KIN-19 and RHO-1::Venus display a significantly decreased fluorescence lifetime compared to Venus only control worms (*Figure 2A–C*, *Figure 2—figure supplement 3A*). Areas with the strongest drop in fluorescence lifetime co-localize with fluorescent-labeled aggregates (*Figure 2A*). Of note, we also observed a similar drop in the fluorescence lifetime of pharyngeal RHO-1 labeled with tagRFP compared to tagRFP alone (*Figure 2—figure supplement 3B*). In both models with KIN-19 and RHO-1, we observed a significant drop in the fluorescence lifetime already at day 1, consistent with the appearance of aggregates in young animals due to protein overexpression. In particular, between day 1 and day 7 of adulthood, we measured a dramatic decrease in fluorescence lifetime of RHO-1 aggregates and a more modest decrease in the fluorescence lifetime of KIN-19 aggregates. This result is consistent with the age-dependent increase in aggregate formation by both proteins and the fact that RHO-1 puncta tend to be larger and more solid than KIN-19 puncta. Indeed, all RHO-1 puncta evaluated showed no recovery after photobleaching, whereas 30% of KIN-19 puncta showed some recovery as previously described (*Figure 2—figure supplement 2C–E* compared to *David et al., 2010*). In addition to evaluating the fluorescence lifetime, we assessed whether RHO-1 aggregates are stained by the Congo red derivative X34 previously used to detect amyloid deposits (*Styren et al., 2000*). Consistent with the amyloid-like nature of the aggregates, we observed co-localization of X34 with RHO-1 aggregates in vivo (*Figure 2—figure supplement 3C–E*).

To gain more insight into the capacity of RHO-1 to form amyloid-like fibrils, we expressed and purified recombinant RHO-1 (*Figure 2—figure supplement 4A,B*) and aggregated it over time by constant shaking in stabilizing buffer to induce fibrillization at 37°C for a week. We obtained low quantities of RHO-1 fibrils, which we were able to analyze by TEM. Analysis of RHO-1 by TEM revealed fibril-like structures which resemble amyloid fibrils such as formed by huntingtin (Htt) (*Figure 2—figure supplement 4C*) (*Reif et al., 2018*). Indeed, RHO-1 fibrils had a morphology more similar to Htt fibrils which are shorter and more branch-like compared to fibrils formed from other amyloids such as alpha-synuclein, which are long and flexible. Additionally, RHO-1 and Htt fibrils display no twisting which can be observed in alpha-synuclein fibrils, suggesting that the packing of the monomeric structure and the interaction between protofibrils may be different (*Lyubchenko et al., 2012*). Importantly, already at day 1, we found comparable fibrillary structures in extracts from RHO-1::Venus transgenic animals (*Figure 2—figure supplement 4D*). In order to confirm that similar fibril structures can be found in the absence of a fluorescent tag, we have isolated fibrils by affinity purification from day 7 old transgenic worms expressing pharyngeal RHO-1::HisAvi and KIN-19::HisAvi and imaged them by TEM (*Figure 2—figure supplement 4E*). Moreover, we have stained isolated RHO-1::HisAvi day 7 fibrils, RHO-1::tagRFP day 2 fibrils and extracts from non-aggregated control

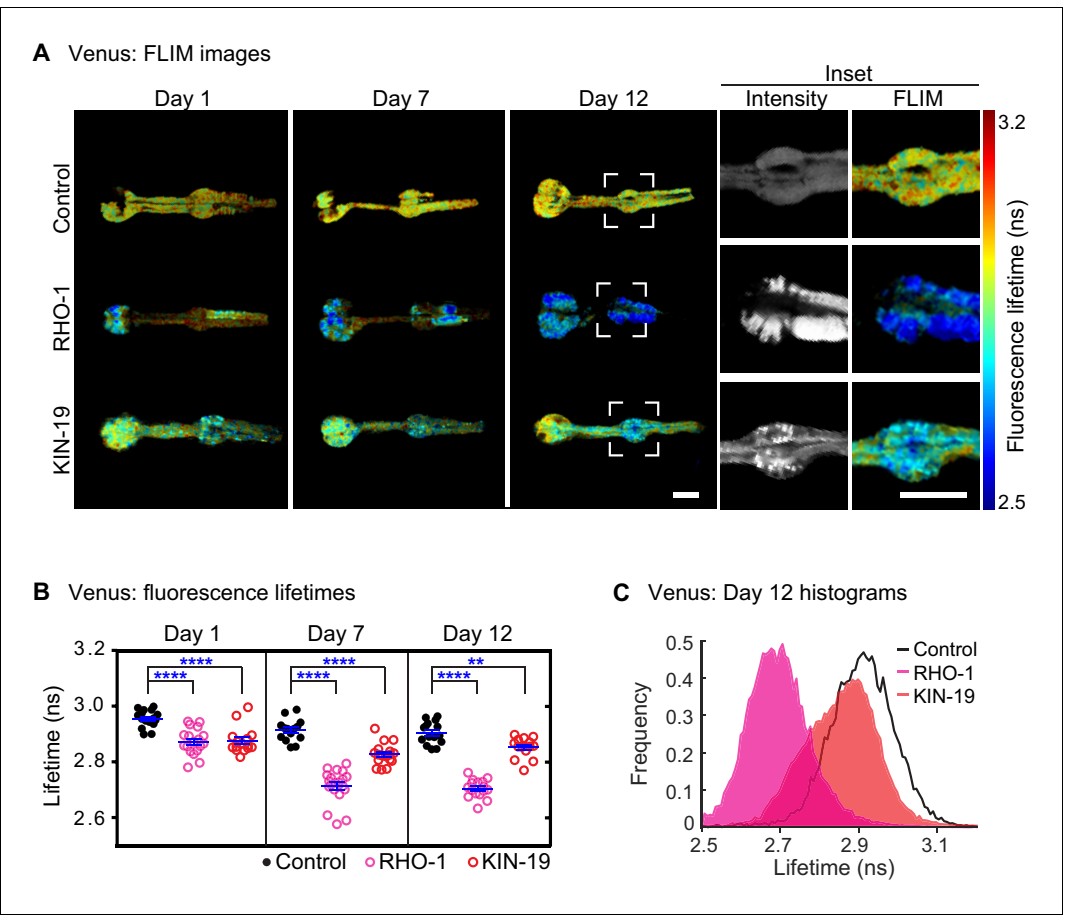

**Figure 2.** Drop in fluorescence lifetime reveals amyloid-like structure of KIN-19 and RHO-1 aggregates in live *C. elegans*. (**A**) Representative intensity-weighted FLIM images of Venus only (control), RHO-1::Venus and KIN-19:: Venus worms. Scale = 25 µm. Inset shows zoomed-in images (intensity and FLIM) of the anterior bulb for Venus only, RHO-1::Venus and KIN-19::Venus worms at day 12. Scale = 25 µm. (**B**) Scatter plot shows a drop in the intensity-weighted fluorescence lifetime averages in adult worms expressing RHO-1::Venus and KIN-19::Venus compared to worms expressing Venus only (control) in the pharynx. n = 7–10, two independent biological repeats. Data are shown as mean lifetime ± SEM and the statistical analysis was performed using two-way ANOVA with Sidak´s multiple comparisons test: **p<0.01, ****p<0.0001. See *Source data 1*. (**C**) Normalized histograms of intensity-weighted fluorescence lifetimes for RHO-1::Venus and KIN-19::Venus show a population shift towards lower lifetimes compared to Venus only (control) worms. The histograms contain information from all pixels of all images acquired for each condition.

DOI: https://doi.org/10.7554/eLife.43059.004

The following figure supplements are available for figure 2:

**Figure supplement 1.** Fibrillation propensity calculate by ZipperDB.
DOI: https://doi.org/10.7554/eLife.43059.005

**Figure supplement 2.** Aggregation with age of RHO-1::Venus and KIN-19::Venus.
DOI: https://doi.org/10.7554/eLife.43059.006

**Figure supplement 3.** Staining of RHO-1::tagRFP with Congo red derivative X34.
DOI: https://doi.org/10.7554/eLife.43059.007

**Figure supplement 4.** RHO-1 forms amyloid-like fibrils in vitro and in vivo.
DOI: https://doi.org/10.7554/eLife.43059.008

**Figure supplement 5.** Structured illumination microscopy reveals that RHO-1 forms ThT-positive fibril structures in vivo.
DOI: https://doi.org/10.7554/eLife.43059.009

worms day 2 with amyloid-binding molecule ThT. Using SIM, we found ThT-positive RHO-1 fibrils as shown previously for polyglutamine and Aβ fibrils in cells (*Lu et al., 2019*; *Young et al., 2016*) (*Figure 2—figure supplement 5*).

Together, these findings strongly suggest that Casein kinase I isoform alpha and Ras-like GTP-binding protein rhoA aggregates contain amyloid-like structures in vivo.

## Age-dependent protein aggregation accelerates functional decline

A reliable indicator of human aging is a decline in physical capacity such as decreased muscle strength and coordination (*Belsky et al., 2015*). Similarly, *C. elegans* displays an age-related decline in pharyngeal pumping and body movement (*Huang et al., 2004*). To investigate whether age-dependent protein aggregation accelerates functional decline with age, we measured pharyngeal pumping and swimming in liquid (thrashing) in transgenic animals overexpressing fluorescent-labeled KIN-19 and RHO-1 either in the pharynx or the body-wall muscles. In both models with KIN-19 and in the model with pharyngeal RHO-1, protein aggregation increased with age, as measured by the change from a diffuse distribution to the formation of specific puncta by the fluorescent-tagged proteins (*Figure 3—figure supplement 1*). Among the transgenic models generated, the largest age-dependent changes were observed for *C. elegans* expressing KIN-19::tagRFP in the pharynx (*Figure 3—figure supplement 1A,E*) (*David et al., 2010*). Although the majority of these transgenic animals have no aggregation at day 2, this dramatically changes with age and at day 8, the majority display high levels of KIN-19::tagRFP aggregation. Importantly, pumping frequency was strongly reduced in day 7 aged animals with the highest levels of pharyngeal KIN-19::tagRFP aggregation compared to those with the lowest aggregation levels (*Figure 3A*, *Supplementary file 1*). The detrimental effect of age-dependent protein aggregation was also apparent in animals with KIN-19::tagRFP aggregation in the body-wall muscles, since we observed an earlier decline in swimming frequency associated with KIN-19::tagRFP aggregation in the body-wall muscle (*Figure 3—figure supplement 2A*). This effect was amplified in animals with the highest level of aggregation (*Figure 3B*). Of note, the fluorescent tagRFP expressed alone in the pharynx or in the body-wall muscle did not affect muscle function (*Figure 3—figure supplement 2A, C, D*). Decreased body movement with age is associated with increased sarcopenia characterized by disordered sarcomeres and reduced F-actin filaments (F-actin) (*Baird et al., 2014*; *Herndon et al., 2002*). To evaluate whether the presence of KIN-19 aggregates is linked to muscle damage at the cellular level, we assessed F-actin staining in animals with KIN-19 aggregation in the body-wall muscles. We found that animals with aggregates tended to have higher levels of disrupted sarcomers compared to controls (*Figure 3—figure supplement 3A,B*).

To further show that protein aggregation is the cause of the functional decline in our transgenic animal models rather than other co-occurring aging factors, we examined the effects of protein aggregation in the absence of aging. Most likely due to the high level of overexpression in the pharyngeal muscles, RHO-1::tagRFP aggregated abundantly already in young animals (*Figure 3—figure supplement 1C,E*). We found that these high levels of RHO-1::tagRFP aggregation strongly impaired pharyngeal pumping in young animals (day 2) (*Figure 3C*). Notably, these young animals displayed pumping rates normally observed only in aged animals. Moreover, RHO-1 aggregation disrupted actin filament structure in the pharyngeal muscles (*Figure 3—figure supplement 3C,D*). To exclude that RHO-1 overexpression itself is toxic to *C. elegans*, we examined another transgenic model where RHO-1::Venus is expressed under the strong *C. elegans* body-wall muscle promoter *Punc-54*. These animals have similar levels of transgene expression compared to animals with body-wall muscle KIN-19::tagRFP (*Figure 3—figure supplement 1F*). Yet, whereas KIN-19 aggregates abundantly in the body-wall muscle, RHO-1::Venus hardly aggregates in this tissue (*Figure 3—figure supplement 1D*). Thus, these transgenics are a suitable control to identify RHO-1 toxicity caused by overexpression. Importantly, we did not observe reduced thrashing in animals with body-wall muscle RHO-1, demonstrating that overexpressed of RHO-1 without concomittant aggregation is not toxic (*Figure 3D*). Similarly, the observation that only animals with the highest levels of pharyngeal KIN-19::tagRFP aggregation have functional impairment shows that KIN-19::tagRFP overexpression alone does not cause pumping defects (*Figure 3A* and *Figure 3—figure supplement 2B*). Together, these positive and negative controls reveal that protein aggregation itself is detrimental in *C. elegans*.

Collectively, these findings show that animals with accelerated protein aggregation experience an earlier onset of functional decline in the tissues affected.

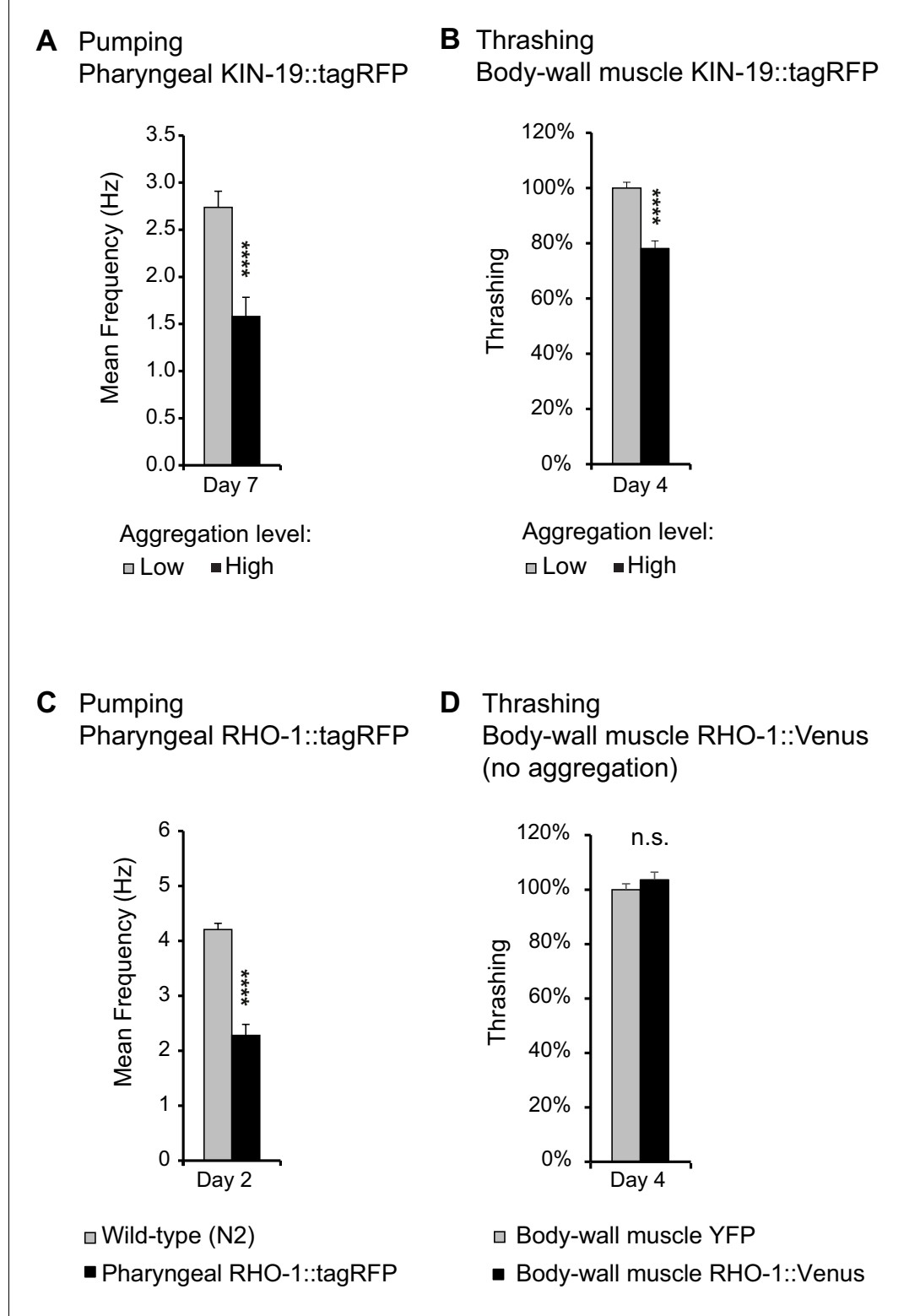

**Figure 3.** Age-dependent protein aggregation impairs pharyngeal and body-wall muscle function. (**A**) Aged animals with high levels of pharyngeal KIN-19 aggregation have reduced pharyngeal pumping. N = 23–28 animals analyzed per group. T-test: p<0.0001. (**B**) Aged animals with high levels of KIN-19 aggregation in the body-wall muscle display reduced thrashing. Mean body bends per seconds are set to 100% in animals with low aggregation. Mann-Whitney test: p<0.0001. (**C**) Young animals with RHO-1 aggregation in pharynx have impaired

*Figure 3 continued on next page*

*Figure 3 continued*

pharyngeal pumping. N = 26–27 animals analyzed per group. T-test: p<0.0001. (D) Overexpression of RHO-1 without aggregation in the body-wall muscles does not influence thrashing. Mean body bends per seconds are set to 100% in *Punc-54::yfp* transgenic animals. Mann-Whitney test: non-significant. SEM represented, independent biological repeats in *Supplementary file 1* and *Source data 1*.

DOI: https://doi.org/10.7554/eLife.43059.010

The following figure supplements are available for figure 3:

**Figure supplement 1.** Markers for age-dependent protein aggregation.
DOI: https://doi.org/10.7554/eLife.43059.011

**Figure supplement 2.** KIN-19 aggregation in body-wall muscles impairs thrashing.
DOI: https://doi.org/10.7554/eLife.43059.012

**Figure supplement 3.** KIN-19 and RHO-1 aggregation impacts muscle structure.
DOI: https://doi.org/10.7554/eLife.43059.013

## Discussion

Widespread protein aggregation in the context of normal aging has been observed in *C. elegans* (*David et al., 2010*; *Reis-Rodrigues et al., 2012*; *Walther et al., 2015*), Drosophila (*Demontis and Perrimon, 2010*), *Saccharomyces cerevisiae* (*Peters et al., 2012*) and in mammals, notably in neural stem cells (*Leeman et al., 2018*), heart (*Ayyadevara et al., 2016b*) and skeletal muscles (*Ayyadevara et al., 2016a*), bone marrow and spleen (*Tanase et al., 2016*). Although these hundreds of proteins are maintained in a functional and soluble state in young animals, they lose their functional structure with age and accumulate into insoluble aggregates. The detergent insoluble properties and solid nature of the aggregates indicate similarities with disease-associated protein aggregation. However, until now it has not been known whether age-dependent protein aggregates display amyloid-like structures, a key characteristic of disease-associated protein aggregation. In this study, we focus on two normally globular proteins, Casein kinase I isoform alpha and Ras-like GTP-binding protein rhoA. We show that aggregates of either proteins display distinct fluorescence quenching properties characteristic of amyloid structures. In neurodegenerative diseases and amyloidosis, protein aggregation is a crucial part of the pathological process. We demonstrate that aggregates formed by proteins prone to aggregation with age contribute to functional decline in the tissues affected. Our results predict that even if only a proportion of the hundreds of proteins becoming insoluble with age form harmful amyloid-like aggregates, this would be a significant cause of tissue aging for a variety of organs.

The speed of aggregate formation and the presence of amyloid-like structure brings significant insight into the aggregation process occurring during normal aging. We demonstrate that newly synthesized proteins rapidly assemble into large aggregates. The recruitment of newly synthesized proteins into existing aggregates indicates a seeding effect. However, we also observed the rapid formation of new aggregates entirely made up of newly synthesized proteins. This result shows that molecular aging of the protein caused by progressive accumulation of damage is not required for proteins to aggregate with age. Moreover, amyloid-like aggregates forming already in young animals indicate an intrinsic aggregation propensity. Therefore, it is likely that these proteins self-assemble into amyloid structures directly from unfolded or partially folded states occurring during or shortly after translation rather than undergoing unfolding from their natively folded state. This conclusion is consistent with the view that protein folding intermediates are particularly at risk of aggregating as exemplified by a recent study revealing that newly synthesized proteins constitute the majority of the insoluble fraction prompted by thermal stress (*Xu et al., 2016*). Furthermore, artificially increasing ribosome pausing during translation causes widespread protein aggregation (*Nedialkova and Leidel, 2015*). Both Casein kinase I isoform alpha and Ras-like GTP-binding protein rhoA contain several hexapeptides with high propensity for fibrillation which are normally buried in the fully folded protein (*Figure 2—figure supplement 1*). Interestingly, the most prominent amyloid-promoting sequence for both proteins is localized near the N-terminus. Therefore, aberrant interactions may start already during translation. As protein aggregation typically results in loss of function, there is a strong evolutionary pressure to avoid this and molecular chaperones have evolved to recognize specifically sequences of high aggregation propensity (*Rousseau et al., 2006*).

Our findings predict that age-dependent protein aggregation would result from decreased levels of molecular chaperones linked to protein synthesis rather than molecular chaperones induced by stress (*Albanèse et al., 2006*; *Pechmann et al., 2013*). Impaired proteasome-mediated removal of unfolded proteins directly after synthesis (*Schubert et al., 2000*) could also significantly contribute to age-dependent protein aggregation. Interestingly, the relative lack of Ras-like GTP-binding protein rhoA aggregation in the body-wall muscles compared to the pharyngeal muscles shows that this protein only aggregates within a specific cellular environment. Thus, tissue-specific factors could be crucial for the aggregation process. These could be differences in the tissue proteostasis network and reliance on certain proteostasis components with age (*Hamer et al., 2010*; *Kern et al., 2010*; *Sala et al., 2017*) but also age-related changes in the local tissue environment. For example, tissue-specific changes in ATP levels (*Patel et al., 2017*), availability of certain ions such as $Ca^{2+}$ (*Lautenschläger et al., 2018*), redox state (*Kirstein et al., 2015*), could explain why certain proteins aggregate in one tissue but not another.

The resemblance between age-dependent protein aggregation and disease-associated protein aggregation raises the question whether they have similar proteotoxic mechanisms. Our results show that there is a strong correlation between the presence of large aggregates and impaired tissue function. In disease, intracellular aggregate toxicity is caused, at least in part, by sequestration and the resulting loss of function of essential cellular proteins (*Hosp et al., 2017*; *Olzscha et al., 2011*; *Yu et al., 2014*). Notably, proteostasis network components are significantly enriched in the age-dependent insoluble proteome (*David et al., 2010*). Therefore, loss-of-function of proteins responsible for preventing aggregation could be a key source of toxicity. The correlation between large aggregates and proteotoxicity observed in the present study, does not exclude that intermediate forms on the path to age-dependent protein aggregation are also proteotoxic. Indeed, disease-associated pre-fibrillar species or oligomers can be highly toxic by interacting through their hydrophobic side chains with other cellular components and in particular lipid membranes (*Fusco et al., 2017*; *Knowles et al., 2014*; *Walsh et al., 2002*). Further experiments will be needed to characterize which types of intermediate species occur during age-dependent protein aggregation and to evaluate their potential toxicity. Conversely, it remains possible that some forms of age-dependent protein aggregation could be protective or manipulated to form harmless non-amyloid aggregates. Finally, it is possible that age-dependent protein aggregation occurring in one tissue will induce accelerated aging in another tissue. Following the evidence for seeding, self-propagation and cell-to-cell transfer of amyloid aggregating species in a disease context (*Jucker and Walker, 2013*), it is intriguing to speculate that the same mechanisms may arise with proteins aggregating during normal aging.

The proteotoxicity and amyloid conformation of age-dependent protein aggregation has important implications for diseases associated with protein aggregation. Accelerated functional decline and the overload of the proteostasis network caused by age-dependent protein aggregation could indirectly enhance disease-associated pathogenesis. There is also evidence for a direct connection between disease and aging-related aggregation. Indeed, a significant proportion of the proteins sequestered in disease pathological deposits are prone to aggregate with age (*Ciryam et al., 2013*; *David et al., 2010*). Notably, Casein kinase I isoform alpha is present in tau aggregates (*Kannanayakal et al., 2008*; *Kuret et al., 1997*). Therefore, age-dependent aggregation-prone proteins interact with disease-aggregating proteins in humans. Furthermore, we have demonstrated that minute amounts of insoluble proteins from aged wild-type mouse brains or aged *C. elegans* are sufficient to cross-seed amyloid-β aggregation in vitro (*Groh et al., 2017a*). One possibility is that the highly hydrophobic nature of the amyloid-like structures in age-dependent protein aggregates provides a destabilizing surface that can promote the conformational conversion of disease-associated aggregating proteins. If the composition of the insoluble proteome is cell-specific, increases in age-dependent protein aggregation, for example through higher expression or somatic aggregation-promoting mutations (*Freer et al., 2016*; *Lodato et al., 2018*), could sensitize specific cells to disease-associated pathogenesis.

In summary, our study demonstrates that aggregation of proteins during normal aging resembles pathological protein aggregation observed in neurodegenerative diseases and amyloidosis. We then show that age-dependent protein aggregation causes early functional decline, a read-out of accelerated aging. These findings emphasize age-dependent amyloid-like aggregation as an important target to restore physical capacity and to promote healthy aging. Already promising results reveal that

lysosome activation in the germline and in aged neural stem cells clears protein aggregates and rejuvenates the cells (*Bohnert and Kenyon, 2017*; *Leeman et al., 2018*).

# Materials and methods

## Key resources table

| Reagent type (species) or resource | Designation | Source or reference | Identifiers | Additional information |
|---|---|---|---|---|
| Gene (*C. elegans*) | *kin-19* | Wormbase | WBGene00002202 | |
| Gene (*C. elegans*) | *rho-1* | Wormbase | WBGene00004357 | |
| Strain (*E. coli*) | OP50-1 | Caenorhabditis genetic center (CGC) | RRID:WB-STRAIN:OP50-1 | Streptomycin resistant strain of OP50 |
| Genetic reagent (*C. elegans*) | *muEx473[Pkin-19::kin-19::tagrfp + Ptph-1::gfp]* | *David et al., 2010* | RRID:WB-STRAIN:CF3166 | N2 background |
| Genetic reagent (*C. elegans*) | *muEx512[Pkin-19::tagRFP + Ptph-1::GFP]* | *David et al., 2010* | CF3317 | N2 background |
| Genetic reagent (*C. elegans*) | *muIs209[Pmyo-3::kin-19::tagrfp + Ptph-1::gfp]* | *David et al., 2010* | CF3649 | N2 background |
| Genetic reagent (*C. elegans*) | *muIs210[Pmyo-3::tagrfp + Ptph-1::GFP]* | *David et al., 2010* | CF3650 | N2 background |
| Genetic reagent (*C. elegans*) | *muEx587[Pkin-19::kin-19::meos2 + Punc-122::gfp]* | This paper | CF3706 | N2 background |
| Genetic reagent (*C. elegans*) | *uqIs9[Pmyo-2::rho-1::tagrfp + Ptph-1::gfp]* | This paper | DCD13 | N2 background |
| Genetic reagent (*C. elegans*) | *uqEx4[Pmyo-3::kin-19::meos2]* | This paper | DCD69 | N2 background |
| Genetic reagent (*C. elegans*) | *uqEx[Pmyo-2::rho-1::meos2 + Punc-122::gfp + cb-unc-119(+)]* | This paper | DCD83 | injected into EG6699 containing ttTi5605II; unc-119(ed3)III |
| Genetic reagent (*C. elegans*) | *uqEx22[Punc-54::rho-1::venus]* | This paper | DCD127 | N2 background |
| Genetic reagent (*C. elegans*) | *uqIs12[Pmyo-2::rho-1::venus]* | This paper | DCD146 | N2 background |
| Genetic reagent (*C. elegans*) | *uqEx37[Pkin-19::kin-19::venus + Punc-122::gfp]* | *Lechler et al., 2017* | DCD179 | N2 background |
| Genetic reagent (*C. elegans*) | *uqIs22 [Pkin19::kin19::hisavi + Pkin19::birAtagrfp]* | This paper | DCD242 | fem-1(hc17ts) IV background |
| Genetic reagent (*C. elegans*) | *uqIs19 [Pmyo2::rho1::hisavi + Pmyo2::birAtagrfp]* | This paper | DCD243 | fem-1(hc17ts) IV background |
| Genetic reagent (*C. elegans*) | *uqEx49[Pkin-19::meos]* | This paper | DCD245 | N2 background |
| Genetic reagent (*C. elegans*) | *uqEx51[Pmyo-2::venus]* | This paper | DCD248 | N2 background |
| Genetic reagent (*C. elegans*) | *rmIs126[Punc-54::YFP]* | https://doi.org/10.1073/pnas.152161099 | RRID:WB-STRAIN:AM134 | N2 background |
| Genetic reagent (*C. elegans*) | *fem-1(hc17ts)IV* | CGC | WB Cat# BA17, RRID:WB-STRAIN:BA17 | |
| Recombinant DNA reagent | tagRFP | Evrogen, pTagRFP-N, FP142 | | |
| Recombinant DNA reagent | pKA1062 containing mEOS2 | Other | | Kaveh Ashrafi, UCSF, USA |
| Antibody | Phalloidin-iFluor 488 conjugate | ABD-23115, AAT Bioquest, Biomol, Germany | | '1:50' |

*Continued on next page*

*Continued*

| Reagent type (species) or resource | Designation | Source or reference | Identifiers | Additional information |
|---|---|---|---|---|
| Chemical compound, drug | X34 | SML1954, Sigma-Aldrich, Germany | | 1 mM final |
| Chemical compound, drug | thioflavin T | #ab120751, abcam, UK | | 50 µM |
| Commercial assay or kit | 5-Hydroxytryptamine creatinine sulfate complex | H7752, Sigma-Aldrich, Germany | | 10 µM |
| Commercial assay or kit | Nickel Sepharose 6 Fast Flow beads from HisTrap FF Crude column | GE Healthcare, Uppsala, Sweden | | |

## Cloning and strain generation

Cloning was carried out using the Gateway system (Life Technologies, Darmstadt, Germany). Pmyo-2 promoter and pKA1062 mEOS2 translational vector were kindly provided by Dr. Brian Lee and Dr. Kaveh Ashrafi, UCSF. *rho-1* cDNA was amplified from a cDNA library prepared from total RNA isolated from N2 worms. Plasmid containing biotinylation enzyme birA was kindly provided by Dr. Ekkehard Schulze (University Freiburg). All constructs contain the unc-54 3′ UTR. The tagrfp vector was obtained from Evrogen (AXXORA, San Diego, CA). Venus was generated by targeted mutation of the yfp gene. HisAvi-tagged KIN-19 and RHO-1 were generated by cloning at the C-terminus a RGSH6 tag together with a bacterially derived polypeptide serving as a biotinylation signal in vivo as previously described (*Schäffer et al., 2010*; *Tagwerker et al., 2006*). Constructs were sequenced at each step. Transgenics were generated by microinjection of the constructs at concentrations between 10 and 50 ng/µl into N2 animals. Stable lines were generated by irradiating the animals containing the extrachromosomal array in a CL-1000 Ultraviolet Crosslinker (UVP) with 275µJ x 100. 100% transmission lines were backcrossed at least four times into the wild-type N2 strain.

## Maintenance

All strains were kept at 15°C on NGM plates inoculated with OP50 using standard techniques. Age-synchronization was achieved by transferring adults of the desired strain to 20°C and selecting their progeny at L4 stage. All experiments were performed at 20°C. Day 1 of adulthood starts 24 hr after L4.

## Photoconversion of mEOS2-tag and quantification of fluorescence levels

For photoconversion, worms were transferred onto a small (diameter 35 mm) NGM plate without food. The plate was placed 0.5 cm below a collimator (Collimator High-End Lumencor, Leica, Germany) fitted with a filter for blue fluorescence (387/11 BrightLine HC, diameter 40 mm) and illuminated by a Lumencor Sola SE II (AHF, Tübingen). Conversion of mEOS2 in transgenic animals was performed four times for five minutes, with 2 min pauses between exposures. To reduce translation, worms were placed 2 hr before conversion on bacterial seeded plates with 500 µg/ml cycloheximide and kept after conversion on plates with cycloheximide for 48 hr during aggregation quantification.

## Aggregation quantification in vivo

Aggregation levels were determined using Leica fluorescence microscope M165 FC with a Planapo 2.0x objective. Aggregation was quantified following pre-set criteria adapted to the transgene expression pattern and levels in the different transgenic *C. elegans* models: Animals expressing *Pkin-19::KIN-19::mEOS2, Pkin-19::KIN-19::Venus* or *Pkin-19::KIN-19::TagRFP* were divided into less than 10 puncta (low aggregation), between 10 and 100 puncta (medium aggregation) and over 100 puncta in the anterior pharyngeal bulb (high aggregation). Animals overexpressing *Pmyo-2::RHO-1:: Venus* were divided into less than 10 puncta in anterior or posterior pharyngeal bulb (low aggregation), over 10 puncta in either bulbs (medium aggregation) and over 10 puncta in both bulbs (high aggregation). Because of extensive RHO-1 aggregation in animals overexpressing *Pmyo-2::RHO-1::*

*TagRFP*, aggregation was only quantified in the isthmus: animals with no aggregation (low aggregation), animals with aggregation in up to 50% (medium aggregation) and animals with aggregation in more than 50% (high aggregation) of the isthmus. Animals overexpressing *Pmyo-3::KIN-19::TagRFP* were divided into over 15 puncta in the head or the middle body region (low aggregation), over 15 puncta in the head and the middle body region (medium aggregation) and over 15 puncta in head, middle body and tail region (high aggregation). The same categories defined for animals overexpressing *Pmyo-3::KIN-19::TagRFP* were used to evaluate animals overexpressing *Pmyo-3::KIN-19:: mEOS2* with a cutoff of 10 puncta instead of 15 to account for slightly lower aggregation levels. Animals overexpressing *Punc-54::RHO-1::Venus* were divided into two categories: less than 15 puncta in the whole animal (low aggregation) and over 15 puncta in the whole animal (medium aggregation). Counting was done in a blind fashion. Two-tailed Fisher's exact test using an online tool (https://www.socscistatistics.com/tests/fisher/default2.aspx) was performed for statistical analysis.

## Confocal imaging

For confocal analysis using a Leica SP8 confocal microscope with the HC PL APO CS2 63x/1.30 NA glycerol objective, worms were mounted onto slides with 2% agarose pads using 2 µM levamisole for anesthesia. Worms were examined using the Leica HyD hybrid detector. The tag mEOS2 was detected using 506 nm as excitation and an emission range from 508 to 525 nm for green fluorescence and 571 nm as excitation and an emission range from 573 to 602 nm for red fluorescence. 3D reconstructions were performed using the Leica Application Suite (LAS X). For X34 imaging, X34 was excited with a 405 nm laser and detected with an emission window between 470 and 520 nm and RHO-1::tagRFP using 555 nm as excitation and an emission range from 565 nm to 620nm. For muscle structure imaging, phalloidin was visualized by excitation at 488 nm and with an emission window between 506 and 551 nm and KIN-19::tagRFP, RHO-1::tagRFP and tagRFP were visualized by excitation at 555 nm and with an emission window between 560 and 650 nm.

FRAP analysis was performed as previously described (*David et al., 2010*) using the Leica SP8 confocal microscope PMT detector. Relative fluorescence intensity (RFI) was analyzed as described previously following the equation RFI = (Tt/Ct)/(T0/C0), where T0 is the intensity in the region of interest (ROI) before photobleaching; Tt, the intensity in the ROI at a defined time after photobleaching; C0, the intensity in the non-bleached part of the puncta before photobleaching; and Ct, the intensity in the non-bleached part of the puncta after bleaching (*Brignull et al., 2006*).

## Fluorescence lifetime imaging in vivo

For fluorescence lifetime imaging, transgenic *C. elegans* were mounted on microscope slides with 2.5% agarose pads using 25 mM NaN3 as anaesthetic. All samples were assayed on a modified confocal-based platform (Olympus FV300-IX70) equipped with a 60x oil objective (PLAPON 60XOSC2 1.4NA, Olympus, Germany) and integrated with time-correlated single photon counting (TCSPC) FLIM implementation. A pulsed supercontinuum (WL-SC-400–15, Fianium Ltd., UK) at 40MHz repetition rate served as the excitation source. YFP was excited at 510 nm using a tuneable filter (AOTFnC-400.650, Quanta Tech, New York). The excitation light was filtered with FF03-510/20 and the fluorescence emission was filtered with FF01-542/27 (both bandpass filters from Semrock Inc, New York) before reaching the photomultiplier tube (PMC-100, Becker and Hickl GmBH, Berlin, Germany). Photons were recorded by a SPC-830 (Becker and Hickl GmBH, Germany) module that permits sorting photons from each pixel into a histogram according to the photon arrival times. Photons were acquired for two minutes to make a single $256 \times 256$ FLIM image and photobleaching was verified to be negligible during this time. Photon count rates were always kept below 1% of the laser repetition rate to avoid pulse pileup. All raw FLIM images were fitted with a single exponential decay function using FLIMfit (*Warren et al., 2013*) and exported to MATLAB (Mathworks, Inc, Natick, MA) to obtain an intensity weighted lifetime average for each image. Statistical analysis was carried out using two-way ANOVA followed by Sidak's multiple comparisons test in Graphpad Prism software (La Jolla, CA).

## X34 staining

Worms were incubated in 1 mM X-34 in 10 mM Tris-HCl pH 8 for 2 hr, gently shaking at room temperature as previously described (*Link et al., 2001*). Worms were then transferred to bacteria seeded NGM plates to destain overnight before confocal imaging.

## Pharyngeal pumping analysis

Electrical activity of the pharyngeal pumping was measured using the NemaMetrix ScreenChip System (NemaMetrix, Eugene, OR). The entire setup is housed in a laboratory that maintained a temperature of approximately 21°C. Baseline noise was typically between 5 and 25 µV.

For each experiment, 50 worms were picked in 1.5 ml of M9 +0.01% Triton and washed three times via low-speed centrifugation. Worms were resuspended in 1.5 ml M9 +0.01% Triton + 10 µM 5-Hydroxytryptamine creatinine sulfate complex (Serotonin creatinine sulfate monohydrate) (Sigma, H7752) and incubated for 20 min. The ScreenChip system was placed on a stereoscope and loaded with a fresh screen chip. The screen chip was then vacuum-filled with M9 +0.01% Triton+10 µM 5-Hydroxytryptamine creatinine sulfate complex and the NemAquire software initiated for baseline noise checking. The animals were loaded into the recording channel of the screen chip via vacuum. After loading each animal, we waited at least 30 s or until the pumping became regular before starting to record. Each animal was recorded for approximately 2 min regardless of whether pumping activity was observed or not. Between 20 and 40 animals were recorded for each condition.

The recordings were analyzed by NemAnalysis v0.2 software using the 'Brute Force' optimization method. The ideal settings were chosen automatically from all combinations of the bounds settings (Minimum SNR from 1.4 (low) to 2.0 (high), with a Step size of 0.1; Highpass Cutoff from 10 (low) to 20 (high), with a Step size of 5) and applied to produce the analysis results. Data was exported into Excel for statistical analysis with the student's t test.

## Thrashing analysis

To quantify movement in terms of body-bends-per-second, movies of worms swimming in liquid were acquired with high frame rates (15 frames per second) using a high-resolution monochrome camera (JAI BM-500 GE, Stemmer imaging GmbH, Puchheim, Germany). For each condition, around 40 animals were filmed (see *Supplementary file 1* for exact numbers). Worms were picked from cultivation plate and allowed to swim in a small plastic petri dish filled with M9 +0.01% Triton. Petri dish containing worms was placed on a transparent platform and illuminated from bottom up with a flat backlight (CCS TH-211/200-RD, Stemmer imaging GmbH, Puchheim, Germany) to achieve homogeneous, high-contrast lighting. The entire setup is housed in a laboratory that maintained a temperature of approximately 21°C. Movies were taken 10 min after placing the animals in the liquid. Five consecutive 30 s movies were made for each group of worms. The movies were then analyzed using the ImageJ wrMTrck plugin (*Nussbaum-Krammer et al., 2015*). The wrMTrck plugin tracked individual worms in the movies and counted the numbers of body-bends. The input values of wrMTrck_Batch are detailed in *Supplementary file 2*. Mann-Whitney test was used for statistical analysis (GraphPad Prism 7).

## Analysis of muscle structure

To analyze the muscle structures in the presence of aggregates, worms were collected at day 4 (for body-wall muscle KIN-19::tagRFP strain CF3649 and tagRFP control strain CF3650) and at day 2 (for pharyngeal RHO-1::tagrfp strain DCD13 and tagRFP control strain CF3317) and fixed in 4% PFA for 10 min at room temperature (RT). Worms were stained with phalloidin to visualize F-actin following a modified protocol by *Karady et al. (2013)*. Briefly, worms were washed in phosphate buffered saline (PBS), incubated for 30 min in PBS with 2% Tween and reduced for 30 min in Tris-Triton β-mercaptoethanol solution (5% β-mercaptoethanol, 1% triton X-100, 130 mM Tris pH 6.8). After washing with PBS, worms were stained with phalloidin (1:50 in PBS, 0.5% triton X-100; Phalloidin-iFluor 488 conjugate from AAT Bioquest), washed and mounted on slides for confocal imaging.

Image analysis of muscle structure was performed in a blind fashion. The body-wall muscle structure was considered normal when the actin filaments are smooth and tightly arranged with no empty space between, modest defective when some of the filaments were slightly distorted or less densely packed. Finally, body-wall muscles were considered severe defective when the filaments appeared

significantly thinner or highly wrinkled, or with large empty space between. The pharyngeal muscle structure was considered severely defective when the actin filament structure showed large holes.

## Plasmid generation for RHO-1 recombinant expression

*C. elegans* RHO-1 cDNA was cloned into pET32a expression vector using restriction sites BamHI and HindIII (NEB, UK). The open-reading frame encoded the RHO-1 fusion protein comprising of a thioredoxin protein, 6xHis tag and the Tobacco etch virus (TEV) cleavage recognition site with the sequence ENLYFQA, where TEV cleaves between Q and A, which was also the N-terminal residue of the RHO-1 protein sequence, followed by the RHO-1 protein. The plasmid was confirmed by DNA sequencing (Source Bioscience, Cambridge, UK).

## Expression and isolation of recombinant RHO-1 from inclusion bodies

Reagents were purchased from Sigma-Aldrich, UK unless stated. BL21 DE3 STAR E. coli (Thermo Fisher Scientific, USA) were transformed with pET32a:RHO-1. 3 l cultures of *E. coli* in Lysogeny Broth containing carbenicillin (100 μg/ml) were grown at 37°C at 250 rpm and induced for expression of the RHO-1 fusion protein with 1 mM isopropyl-β-thiogalactopyranoside (IPTG) for 4 hr. *E. coli* were pelleted by centrifuge at 8000 x g for 15 min before being washed by resuspension in PBS with 1% Triton X-110 and 1 mM Tris(2-carboxyethyl)phosphine hydrochloride (TCEP) and centrifuged again. The pelleted *E. coli* were either stored at −20°C until further use or lysed straight away. The RHO-1 fusion protein forms in inclusion bodies. To release the inclusion bodies, the *E. coli* were resuspended in 20 ml of lysis buffer per 1 l of culture (50 mM Tris, 500 mM NaCl, 5 mM $MgCl_2$, 1 mM phenylmethylsulfonyl fluoride (PMSF), protease inhibitor tablets (cOmplete, Mini EDTA-free, Roche), 1% Triton X-110, 1 mM TCEP, pH 8 at 4°C) and sonicated on ice using three rounds of 30 s on and 30 s off 70% sonication power. The suspension was centrifuged at 10,000 x g for 10 min at 4°C. The pellet containing RHO-1 inclusion bodies was resuspended in 30 ml per 1 l culture of wash buffer 1 (Lysis buffer +2 M urea, pH 8 at 4°C). The inclusion bodies were sonicated for four rounds of 10 s on, 20 s off and centrifuged at 10,000 x g for 10 min at 4°C, this wash was then repeated again. Wash numbers 3 and 4 used the same sonication and centrifuge parameters, but pellets were washed with wash buffer 2 (Lysis buffer, 2 M urea, 1 mM TCEP, without Triton X-110 or protease inhibitors). The final pellet became paler and more chalk-like. These wash steps are important to lead to a purer final RHO-1 protein. The inclusion bodies were then solublized in solublizing buffer using 10 ml per 1 l of culture for 1 hr using a magnetic stirrer (50 mM Tris, 500 mM NaCl, 5 mM $MgCl_2$, 6 M guanidinium hydrochloride (GuHCl), 1 mM TCEP, (6 M Urea can also be used in place of GuHCl, but GuHCl gives a slightly higher final protein yield)). Insoluble material was removed by centrifuge at 16,000 x g for 10 min at room temperature (RT).

## Purification of recombinant RHO-1

The RHO-1 fusion protein was purified using two linked-together 1 mL HisTrap Crude FF columns on an ÄKTA Pure (GE Healthcare, Sweden). The columns were equilibrated with solublizing buffer before 10 ml of protein was loaded onto the column by the sample pump. The columns were washed in wash buffer (50 mM Tris, 500 mM NaCl, 5 mM $MgCl_2$, 6 M Urea, 20 mM imidazole) before being eluted against a linear gradient of elution buffer (50 mM Tris, 500 mM NaCl, 5 mM $MgCl_2$, 6 M Urea, 500 mM imidazole) over 16 column volumes of a 1 × 1 ml column, that is 16 ml. Multiple purification runs were performed until all RHO-1 fusion protein was purified. Fractions from all runs containing the RHO-1 fusion protein were pooled and dialysed overnight in 2 l stabilizing buffer modified from (*Healthcare GE, 2007*; *Nelson et al., 2014*; *Thomson et al., 2012*) (50 mM Tris, 250 mM NaCl, 100 mM arginine, 5 mM reduced glutathione, 0.5 mM oxidised glutathione, 5 mM MgCl2, 5 μM guanosine 5'-diphosphate (GDP) (Generon, UK) pH 7.2). The RHO-1 fusion protein was then concentrated to 1 ml per 1 l culture using Spectra/Gel Absorbent (Spectrum Labs, USA) through a 10 kDa MWCO Slide-A-Lyzer dialysis cassette (Thermo Fisher Scientific). The RHO-1 fusion protein was then incubated overnight with recombinant TEV protease to cleave the RHO-1 from the fusion tag in a 1:50 ration of TEV to RHO-1 fusion protein based on absorbance at 280 nm. Recombinant TEV was produced by Dr Marielle Wälti using methods from *Tropea et al. (2009)*. The cleaved RHO-1 was separated from the fusion tag and TEV protein by purification using the two linked-together 1 ml HisTrap Crude FF columns on an ÄKTA Pure. The columns were equilibrated in

stabilizing buffer and 1 ml of the protein loaded onto the columns by injection. Cleaved RHO-1 was eluted in stabilization buffer during the column wash. The fusion tag and the His-tagged TEV were retained on the column and eluted with 100% elution buffer over 10 CV (stabilization buffer with 500 mM imidazole). Protein concentration was monitored throughout purification using the extinction coefficient 0.862 $M^{-1}$ $cm^{-1}$ for the fusion protein, 0.849 $M^{-1}$ $cm^{-1}$ for the refolded fusion protein and 0.875 $M^{-1}$ $cm^{-1}$ for the cleaved refolded RHO-1 protein. Quantitative analysis of protein purity was performed in FIJI image analysis software (*Schindelin et al., 2012*) by profiling protein band intensity of the stained gel and mass spectrometry confirmed the purification of RHO-1. To note, RHO-1 precipitated in NaP, Tris, NaCl buffers.

## Fibrillization of RHO-1 and analysis by transmission electron microscopy (TEM)

RHO-1 was fibrillized by incubating 20 µM in stabilizing buffer during a Thioflavin-T (ThT) based-assay with 10 µM ThT (AbCam, UK) in non-binding, clear bottom, black 96-well plate (PN 655906 Greiner Bio-One GmbH, Germany). The plate was incubated at 37°C with constant shaking at 300 rpm for 7 days. RHO-1 was taken from the wells in the microplate for imaging by TEM. All samples also contained 0.05% $NaN_3$ to prevent bacterial growth.

Fibrillized RHO-1 samples were centrifuged for 20 min at 21,000 x g and the supernatant removed leaving 10 µl of sample. Each 10 µl sample was incubated on a glow-discharged copper grid for 1 min. Excess liquid was blotted off and the grid washed in twice in dH2O for 15 s. 2% uranyl acetate was used to negatively stain the samples for 30 s before imaging on the Tecnai G2 80-200kv TEM at the Cambridge Advanced Imaging Centre.

## Preparation and analysis of worm lysates for fibrils by TEM

*C. elegans* DCD146 expressing RHO-1::Venus, DCD242 expressing KIN-19::HisAvi and DCD243 expressing RHO-1::HisAvi were grown to confluency on high growth medium plates and bleached to obtain a synchronized population of worms as previously described (*Sulston, 1988*). L1s were transferred into a liquid culture with complete S basal supplemented with OP50-1 (OP50 with Streptomycin resistance) and grown at 20°C or 25°C (to induce sterility of DCD242 and DCD243) as previously described (*Groh et al., 2017a*; *Groh et al., 2017b*). At day 1 of adulthood (DCD146) or at day 7 of adulthood (DCD242 and DCD243), worms were allowed to sediment in a separation funnel and washed with cold M9. The worm pellet was resuspended in PBS with 2x protease inhibitor tablets (cOmplete, Mini EDTA-free, Roche) and frozen in liquid nitrogen. For TEM with RHO-1::Venus, frozen worms were resuspended in radioimmunoprecipitation assay (RIPA) buffer with protease inhibitor tablets and lysed by 20–25 passages through a cell homogeniser (Isobiotec, Germany) using a tungsten carbide ball with 16 µm clearance. Cuticle fragments and unlysed worms were removed by centrifugation for 5 min at 835 g and 4°C. After careful removal of the supernatant, the insoluble fraction was collected by centrifugation for 30 min at 21,000 g and 4°C. The supernatant was removed and the pellet was resuspended in PBS with protease inhibitor tablets by homogenising with a needle (27G, Sterican). To perform TEM with fibrils isolated by affinity, we pulled down His-tagged RHO-1 and KIN-19 by nickel beads. For this, 250 µl worm extract was added to 250 µl RIPA buffer (Invitrogen) with protease inhibitors and sonicated for 10 s twice. Cuticle fragments and unlysed worms were removed by centrifugation for 1 min at 800 x g. The supernatant was removed and centrifuged at 21 k x g for 20 min. The pellet was resuspended in 250 µl PBS with protease inhibitors and passed through a 30 G needle three times. 20 µl of precharged Nickel Sepharose 6 Fast Flow beads taken from a HisTrap FF Crude column (GE Healthcare, Uppsala, Sweden) were incubated with the resuspended pellet overnight at 4°C on a fixed speed rotator at 20 rpm (SB2, Stuart, Staffordshire, UK). To separate the beads from unbound proteins the tubes were centrifuged at 800 x g for 1 min and the supernatant removed. 60 µl of 500 mM imidazole in PBS pH 8.0 was added to the beads and incubated overnight at 4°C on a fixed speed rotator at 20 rpm. To isolate eluted fibrils the tubes were centrifuged at 800 x g for 1 min and the supernatant containing the fibrils was used for TEM experiments. 10 µl sample was applied to a carbon coated grid, and 2% uranyl acetate was used for negative staining. Imaging was performed on the Tecnai G2 80-200kv TEM at the Cambridge Advanced Imaging Centre.

## Preparation and analysis of worm lysates for fibrils by SIM

Nunc Lab-Tek II Chamber Slide (Sigma, Dorset, UK) were coated for 30 min with 0.01% poly-L-Lysine (P4707, Sigma) before incubation for 1 hr with either nickel bead extracted fibrils or resuspended worm pellets prepared as described above. RHO-1::tagRFP expressing transgenics (DCD13) and *fem-1(-)* mutants (CF2137; the non-aggregated control worm) extracts were incubated with 50 µM thioflavin T (ThT) (#ab120751, abcam, UK) for 1 hr and washed three times in PBS before imaging. To visualize amyloids from worm extracts, we used our custom-built SIM providing a spatial resolution approaching 90 nm at frame rates reaching 22 Hz (*Young et al., 2016*). Hardware control and image reconstruction were performed with software written in LabView and Matlab (*Ströhl and Kaminski, 2015*). For visualization, ImageJ was used.

## Acknowledgements

Some strains were provided by the CGC, which is funded by NIH Office of Research Infrastructure Programs (P40 OD010440). We thank Dr. Emily Crawford for generating the gateway vector with the histidine-avidin tag and Katja Widmaier for technical assistance.

## Additional information

### Funding

| Funder | Grant reference number | Author |
|---|---|---|
| Deutsches Zentrum für Neuro-degenerative Erkrankungen | | Della C David |
| European Commission | Marie Curie International Reintegration grant 322120 | Della C David |
| Engineering and Physical Sciences Research Council | | Clemens F Kaminski |
| Wellcome | 203249/Z/16/Z | Gabriele S Kaminski Schierle |
| Medical Research Council | MR/N012453/1 | Gabriele S Kaminski Schierle |
| Alzheimer's Research UK | ARUK-PG2013-14 | Gabriele S Kaminski Schierle |
| Infinitus China Ltd | | Clemens F Kaminski Gabriele S Kaminski Schierle |
| Alzheimer's Research UK | Travel grant | Amberley D Stephens |
| Biotechnology and Biological Sciences Research Council | BB/P027431/1 | Romain F Laine |
| Biotechnology and Biological Sciences Research Council | BB/R021805/1 | Romain F Laine |

The funders had no role in study design, data collection and interpretation, or the decision to submit the work for publication.

### Author contributions

Chaolie Huang, Conceptualization, Formal analysis, Validation, Investigation, Visualization, Methodology, Writing—original draft, Writing—review and editing; Sara Wagner-Valladolid, Validation, Investigation, Visualization, Writing—original draft; Amberley D Stephens, Resources, Formal analysis, Validation, Investigation, Visualization, Methodology, Writing—original draft, Writing—review and editing; Raimund Jung, Resources, Formal analysis, Validation, Investigation, Methodology; Chetan Poudel, Formal analysis, Validation, Investigation, Visualization, Methodology, Writing—review and editing; Tessa Sinnige, Validation, Investigation, Visualization, Methodology, Writing—original draft, Writing—review and editing; Marie C Lechler, Resources, Investigation, Visualization, Writing—original draft; Nicole Schlörit, Resources, Investigation, Methodology; Meng Lu, Investigation, Visualization, Methodology; Romain F Laine, Claire H Michel, Investigation, Methodology; Michele Vendruscolo, Supervision, Writing—review and editing; Clemens F Kaminski, Resources, Funding

acquisition; Gabriele S Kaminski Schierle, Conceptualization, Resources, Supervision, Funding acquisition, Writing—original draft, Project administration, Writing—review and editing; Della C David, Conceptualization, Resources, Formal analysis, Supervision, Funding acquisition, Validation, Investigation, Visualization, Methodology, Writing—original draft, Project administration, Writing—review and editing

### Author ORCIDs

Amberley D Stephens https://orcid.org/0000-0002-7303-6392
Tessa Sinnige https://orcid.org/0000-0002-9353-126X
Meng Lu http://orcid.org/0000-0001-9311-2666
Romain F Laine http://orcid.org/0000-0002-2151-4487
Michele Vendruscolo http://orcid.org/0000-0002-3616-1610
Clemens F Kaminski https://orcid.org/0000-0002-5194-0962
Gabriele S Kaminski Schierle http://orcid.org/0000-0002-1843-2202
Della C David http://orcid.org/0000-0001-8597-9470

### Decision letter and Author response

Decision letter https://doi.org/10.7554/eLife.43059.019
Author response https://doi.org/10.7554/eLife.43059.020

## Additional files

### Supplementary files

• Supplementary file 1. Pharyngeal pumping and thrashing repeats.
DOI: https://doi.org/10.7554/eLife.43059.015

• Supplementary file 2. Parameters for thrashing analysis.
DOI: https://doi.org/10.7554/eLife.43059.016

• Source data 1. Analysis of protein aggregation and measures of muscle function in *C. elegans*.
DOI: https://doi.org/10.7554/eLife.43059.014

• Transparent reporting form
DOI: https://doi.org/10.7554/eLife.43059.017

### Data availability

All data generated or analysed during this study are included in the manuscript and supporting files. Source data files have been provided for Figures 1, 2, 3 and Figure 1-figure supplement 1, Figure 2-figure supplement 2, 3, Figure 3-figure supplement 1, 2, 3.

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
