## [Decision Letter]

Thank you for submitting your article "Intrinsically aggregation-prone proteins form amyloid-like aggregates and contribute to tissue aging in *C. elegans*" for consideration by *eLife*. Your article has been reviewed by three peer reviewers, and the evaluation has been overseen by John Kuriyan as the Reviewing and Senior Editor. The reviewers have opted to remain anonymous.

The reviewers have discussed the reviews with one another and the Reviewing Editor has drafted this decision to help you prepare a revised submission.

Summary

This manuscript builds on a previous observation which established that aggregation of proteins during normal aging is widespread, and it examines how and when aggregation occurs, and whether these aggregates resemble disease-associated amyloids.

The authors use two previously-identified aggregation prone proteins, Casein kinase I isoform alpha (KIN-19) and Ras-like GTP-binding protein rhoA (RHO-1), to address these questions. Specifically, they use labeling with mEOS2, a green-to-red photo-convertible fluorescent protein tag, to demonstrate that newly synthesized proteins rapidly assemble into large insoluble aggregates. They demonstrate the recruitment of newly-synthesized proteins (green) into existing aggregates, suggesting a seeding effect, but also formation of new aggregates de novo. The authors show that protein aggregation is a continuous process in their *C. elegans* model, with a gradual clearance of some "old" aggregates and a gradual (on the timescale of one day) appearance of new aggregates, plus some addition of new protein to existing aggregates. The photoconversion technique (green to red) used by the authors to distinguish new protein molecules from pre-existing protein molecules is quite clever. It may not be surprising that relatively newly synthesized protein molecules can add to existing aggregates, or that newly-synthesized protein molecules can nucleate new aggregates. Nevertheless, it is important to demonstrate these things concretely, as is done here.

There are two major conclusions of this paper. One is that protein quality control can potentially be a contributor in proteostasis collapse early in adulthood. Another conclusion is that wild type aggregation-prone proteins can form amyloid in vivo, and the demonstration of such behavior in proteins not associated with disease suggests that this could be key process in aging. These points are of general interest, and make this paper suitable for *eLife*, in principle.

Please consider the following issues raised by the reviewers in preparing a revised version of the manuscript.

Critical points to address:

1) The conclusion that molecular aging of proteins and aggregation is not necessarily caused by progressive accumulation of damage is weakened by the observation that not all tagged proteins photo-convert. This conclusion would be supported by examining the impact on aggregation of blocking translation for a short time. The fact that after the photo-conversion switch there are green fluorescent proteins in aggregates suggests that these are newly-expressed proteins. The fact that there is no viable red core also suggests that these are newly-expressed proteins. There might, however, be small dimeric/oligomeric seeds that are required, and their assembly might take time. Doing a similar experiment on Day 1 rather than Day 5 – or before there are visible aggregates, or by blocking translation – should provide additional support for the key inference.

2) Many proteins have been shown to form amyloids in vitro. To support the claim that KIN-19 and RHO-1 make amyloid-like aggregates in vivo, it is important to use well-established assays supporting the formation of cross-β structures (such as ThT staining). We are concerned that monitoring fluorescence quenching may not be a strong and direct indication of the specific structural signature of an amyloid.

Other points to address:

1) The authors demonstrate that aggregates load (determine by foci numbers of tagRFP KIN-19 or RHO-1) is associated with accelerated functional decline in the tissues affected, suggesting that this could be a significant cause of tissue aging for a variety of organs (Figure 3 and Figure 3—figure supplement 1 and 2). Please explain why a third model system, consisting of KIN-19::tagRFP and RHO-1::tagRFP, was used to monitor toxicity, and not one of the models presented in Figures 1 and 2.

2) The authors claim that the aggregates load "leads to" functional decline. This claim too strong based on the data presented, and we suggest using the word "associated" rather than "leads to".

3) In Figure 2C, we suggest that the authors show fluorescence lifetime histograms and representative decays, so that the reader can see the length of the time window employed, the background level, the number of counts in the peak and the quality of the decay and also perhaps the fits and residuals.

4) Figures 1A and 1B show the percentage of animals that exhibit detectable green or red protein aggregates, rather than some measure of the average quantity (volume?) of green and red aggregates per animal. Please explain why this was done, or switch the analysis as suggested.

---

## [Author Response]

Critical points to address:1) The conclusion that molecular aging of proteins and aggregation is not necessarily caused by progressive accumulation of damage is weakened by the observation that not all tagged proteins photo-convert. This conclusion would be supported by examining the impact on aggregation of blocking translation for a short time. The fact that after the photo-conversion switch there are green fluorescent proteins in aggregates suggests that these are newly-expressed proteins. The fact that there is no viable red core also suggests that these are newly-expressed proteins. There might, however, be small dimeric/oligomeric seeds that are required, and their assembly might take time. Doing a similar experiment on Day 1 rather than Day 5 – or before there are visible aggregates, or by blocking translation – should provide additional support for the key inference.

To address these concerns, we have evaluated aggregate formation starting the conversion in young animals at day 2 (Figure 1—figure supplement 1E). These data show that newly synthesized KIN-19 can aggregate in young animals (albeit in only a small proportion of the worm population compared to day 5). We have also blocked translation and we observed a strong reduction in the formation of new aggregates both starting the conversion at day 2 or at day 5 (Figure 1B and Figure 1—figure supplement 1E).

*2) Many proteins have been shown to form amyloids* in vitro*. To support the claim that KIN-19 and RHO-1 make amyloid-like aggregates* in vivo*, it is important to use well-established assays supporting the formation of cross-β structures (such as ThT staining). We are concerned that monitoring fluorescence quenching may not be a strong and direct indication of the specific structural signature of an amyloid.*

We have added two additional figures in the supplementary section (Figure 2—figure supplement 3 and 5) to address these concerns. In Figure 2—figure supplement 3, we show that in vivo RHO-1::tagRFP aggregates bind to the Congo red derivative X34. In addition in Figure 2—figure supplement 5, we show that in vivo-formed fibrils which have been extracted from transgenic animals expressing RHO-1::HisAvi (at day 7) and RHO-1::tagRFP (at day 2) are ThT positive using structured illumination microscopy. SIM has a resolution of 100 nm and the images shown are similar to polyglutamine and Abeta fibrils imaged by SIM and published previously by the group (Live-cell super-resolution microscopy reveals a primary role for diffusion in polyglutamine-driven aggresome assembly M Lu, L Banetta, LJ Young, EJ Smith, GP Bates, A Zaccone, GSK Schierle, et al., Journal of Biological Chemistry 294 (1), 257-268 and Structural progression of amyloid-β Arctic mutant aggregation in cells revealed by multiparametric imaging M Lu, N Williamson, A Mishra, CH Michel, CF Kaminski, A Tunnacliffe, et al., Journal of Biological Chemistry 294 (5), 1478-1487). Note, since RHO-1 is more aggregation prone in vivo than in vitro we have not been able to run a conventional ThT aggregation assay in vitro.

Other points to address:1) The authors demonstrate that aggregates load (determine by foci numbers of tagRFP KIN-19 or RHO-1) is associated with accelerated functional decline in the tissues affected, suggesting that this could be a significant cause of tissue aging for a variety of organs (Figure 3 and Figure 3—figure supplement 1 and 2). Please explain why a third model system, consisting of KIN-19::tagRFP and RHO-1::tagRFP, was used to monitor toxicity, and not one of the models presented in Figures 1 and 2.

Our initial models were constructed with the monomeric fluorescent protein “tagRFP” (see David et al., 2010). We evaluated toxicity using these initial models which have been integrated and backcrossed since our publication (except for CF3166). The mEOS tagged models were created afterwards to evaluate aggregation dynamics. The Venus tagged models were specifically created for the purpose of FLIM. We had decided to design these separate Venus-tagged models for two reasons: a) Venus models on amyloid aggregation had been designed and analyzed previously in GSKS’ lab and thus we had an internal control to which we could compare measured fluorescence lifetime changes for RHO-1 and KIN-19 and b) the fluorescence lifetime of Venus is in the 3 ns range compared to RFP (2ns) and thus offers a better dynamic range to study fluorescence lifetime dependent protein aggregation. For this revision though, we have now examined fluorescence lifetime in the RHO-1::tagRFP model used for toxicity and similar to the RHO-1::Venus model, we observed a significant drop compared to tagRFP (Figure 2—figure supplement 3B).

2) The authors claim that the aggregates load "leads to" functional decline. This claim too strong based on the data presented, and we suggest using the word "associated" rather than "leads to".

We have modified the corresponding sentence “We demonstrate that aggregates formed by proteins prone to aggregation with age lead to accelerated functional decline in the tissues affected” to “We demonstrate that aggregates formed by proteins prone to aggregation with age contribute to functional decline in the tissues affected.”

To support our findings, we have added an evaluation of the actin filament structure in the body-wall muscles and pharyngeal muscles in the presence of aggregates (Figure 3—figure supplement 3). We observed a significant correlation between disrupted muscle structure and aggregate load underlining the toxicity of protein aggregates also at a cellular level.

3) In Figure 2C, we suggest that the authors show fluorescence lifetime histograms and representative decays, so that the reader can see the length of the time window employed, the background level, the number of counts in the peak and the quality of the decay and also perhaps the fits and residuals.

This has been added, see new Figure 2 and Figure 2—figure supplement 3A.

4) Figures 1A and 1B show the percentage of animals that exhibit detectable green or red protein aggregates, rather than some measure of the average quantity (volume?) of green and red aggregates per animal. Please explain why this was done, or switch the analysis as suggested.

We need to examine the same animals over time to follow the relative changes in aggregation. This is done in live animals under a low magnification fluorescent microscope. In these settings, the best method for aggregate quantification is to categorize each animal into a predefined group: less than 10 aggregates, between 10 and 100 and a group with over 100 aggregates. In order to measure the exact quantity of aggregates as suggested, we would need high magnification images. Unfortunately, this requires immobilization of the animals on microscope slides and these animals are highly difficult to recover for another imaging session.